# TiO_2_-Doped Chitosan Microspheres Supported on Cellulose Acetate Fibers for Adsorption and Photocatalytic Degradation of Methyl Orange

**DOI:** 10.3390/polym11081293

**Published:** 2019-08-02

**Authors:** Xuejuan Shi, Xiaoxiao Zhang, Liang Ma, Chunhui Xiang, Lili Li

**Affiliations:** 1Key Laboratory of Automobile Materials, Ministry of Education, and College of Materials Science and Engineering, Jilin University, Changchun 130022, China; 2Department of Apparel, Events and Hospitality Management, 31 MacKay Hall, Iowa State University, Ames, IA 50011, USA

**Keywords:** TiO_2_, chitosan, electrospinning, electrospraying, methyl orange, photocatalytic degradation

## Abstract

Chitosan/cellulose acetate (CS/CA) used as a biopolymer systema, with the addition of TiO_2_ as photocatalyst (C-T/CA) were fabricated by alternating electrospinning/electrospraying technology. The uniform dispersion of TiO_2_ and its recovery after the removal of methyl orange (MO) was achieved by incorporating TiO_2_ in CS electrosprayed hemispheres. The effects of pH values, contact time, and the amount of TiO_2_ on adsorption and photocatalytic degradation for MO of the C-T/CA were investigated in detail. When TiO_2_ content was 3 wt %, the highest MO removal amount for fiber membranes (C-T-3/CA) reached 98% at pH value 4 and MO concentration of 40 mg/L. According to the data analysis, the pseudo-second-order kinetic and Freundlich isotherm model were well fitted to kinetic and equilibrium data of MO removal. Especially for C-T-3/CA, the fiber membrane exhibited multiple layers of adsorption. All these results indicated that adsorption caused by electrostatic interaction and photocatalytic degradation were involved in the MO removal process. This work provides a potential method for developing a novel photocatalyst with excellent catalytic activity, adsorbing capability and recycling use.

## 1. Introduction

With the quick progress of industrial technology, water pollution generated from organic dye contaminants has become an increasing global environmental concern in recent years [1,2,3,4]. Organic dyes are mainly generated from manufacturing, such as textile finishing, plastics, paper, cosmetics, pharmaceuticals and food processing. It can cause serious problems for both the environment and to human beings [5,6,7,8,9,10]. Up until now, several methods such as chemical oxidation, coagulation, adsorption, biodegradation, photocatalysis, and reverse osmosis, have been developed to treat dye contaminated wastewater [11,12,13,14]. Particularly, photocatalysis is considered a “green” treatment method which uses solar energy effectively for the degradation of organic pollutants [15,16,17,18,19]. Adsorption is also considered a valid method for the removal of dye in aqueous solutions with high efficiency, low price and easy operation [20,21,22].

At present, researchers have found that the combination of adsorption and photocatalysis had excellent potential applications for dye treatment [23]. TiO_2_ is a type of common semiconductor photocatalyst on account of its prominent characteristics of chemical resistance, excellent catalytic activity, mechanical robustness, and low cost [24,25,26,27,28]. While the low dispersibility, inconvenient recovery, low adsorption performance of TiO_2_ limit its use in photocatalysis applications [29]. Absorbent materials with good adsorption performance on organic pollutants are often used as carrier materials to improve the photocatalysis capability of TiO_2_.

Ideal carrier materials should provide stable anchoring of TiO_2_, outstanding structure stability, selective adsorption for target dye pollutions and the transfer of organic dye molecules to the vicinity of the photocatalytic site. Multifarious materials have been developed for photocatalytic degradation carrier materials, comprising silica, ceramic, polymers, activated carbon, zeolite, and glass. As an environmentally friendly, nontoxic, biodegradable, biocompatible natural macromolecule [30,31,32,33,34], CS with functional groups of hydroxyl and amino groups derived from deacetylation of polysaccharide chitin, is a promising carrier material [35,36,37,38]. Li et al. produced pure CS membranes by electrospinning, exhibiting high adsorption capacity for acid blue-113 [37]. Huang et al. produced an organoclay/CS hybrid system, allowing efficient and concurrent adsorption of different dyes [38]. However, CS is easily soluble in water when amino groups are protonated, making it difficult to maintain integrity of the whole materials. As a hydrophilic, environmentally friendly, biodegradable, and renewable material [39,40,41,42,43], CA has characteristics of non-specific adsorption, and its hydroxyl and ester groups can form stable hydrogen bonding with certain polar functional groups [44,45]. Electrospinning technology is a common and effective method for the preparation of adsorbents [14]. Doh et al. electrospun TiO_2_ nanofibers had a higher degradation rate of organic dye than TiO_2_ nanoparticles because of a high specific surface area and excellent dispersibility [4]. Razzaz et al. proved that CS/TiO_2_ electrospun membranes had higher effective surface area, structural stability and adsorption capacity for metal ions compared with TiO_2_-coated CS nanofibers [46].

In this study, based on CA electrospun fibers, C-T/CA fiber membranes were successfully prepared via electrospraying CS/TiO_2_ concave hemispheres on the surface of CA fibers. As a substrate material, CA was expected to improve the water flux of fiber membranes effectively so that aqueous solutions of organic pollutant, MO could rapidly penetrate into the interior of membranes. The CS hemispheres might disperse TiO_2_ nanoparticles, have good adsorption performance of MO, and transfer MO to the vicinity of TiO_2_ active sites for further photocatalytic degradation. The morphology of fiber membranes were characterized by SEM. Effects of TiO_2_ content, pH values, contact time at different MO concentration for adsorption and photocatalytic degradation of MO were investigated. The MO removal mechanism was surveyed by FTIR, XPS and BET.

## 2. Experiment

### 2.1. Materials

CS (90% deacetylated, average M_w_ = 200,000 Da), TiO_2_ nanoparticles (anatase-type nanopowder, particle size < 25 nm, metal basis), CA (white powder, M_n_ ≈ 30,000 Da, acetyl content ≈ 39.7 wt %, DS = 2.45) and MO were all purchased from Aladdin Industrial Corporation, Shanghai, China. Acetic acid (HAc) and acetone were purchased from Beijing Chemical Works, Beijing, China. Deionized water was obtained from the laboratory. The chemicals were all of analytical grade and used without further purification.

### 2.2. Preparation of Polymer Solutions

10 wt % CA (*w/w*) was dissolved in acetone/deionized water (85/15, *w/w*) and 2 wt % CS (*w/w*) was dissolved in 90% acetic acid (*v/v*), respectively. The solutions were stirred for 24 h at room temperature to obtain the uniform solutions. Then, the various contents of TiO_2_ (1%, 2%, 3%, 4%, 5%, *w/w*) were added to CS solutions at room temperature and stirred for 24 h. The C-T-X solutions were obtained, where X stands for the content of TiO_2_ (1, 2, 3, 4, 5 wt %). The solutions were placed in an ultrasonic water bath for 30 min before using. The frequency of ultrasonication was 25 kHz.

### 2.3. Fabrication of C-T/CA Fiber Membranes

The fabrication processes including (I) electrospinning and (II) electrospraying are shown in Figure 1. The solution was loaded into a 5 mL plastic syringe. The electrospinning conditions included the applied voltage of 20 kV, the 15 cm distance between the tip of the needle and the collector, and the flow rate of 1.0 mL/h. The condition for electrospraying was 18 kV voltage with a distance of 15 cm between the tip of the needle and the collector, and the flow rate was 0.3 mL/h. Then the membrane was dried at 25 °C for 24 h, and peeled off aluminum foil for further study.

### 2.4. Characterizations

The viscosity of precursor solutions was measured by a viscosimeter (NDJ-1, Yutong, Shanghai, China). The electrical conductivity of the solutions was measured using a conductometer (DDS-11A, Shengci, Shanghai, China). All measurements were taken 3 times at room temperature. The morphology and structure of membranes were characterized by SEM (JSM-6700F, JEOL, Tokyo, Japan). All samples were sputtered with platinum under vacuum before assessment. The average microsphere diameters were measured by Image J software. Surface area, pore volume and pore size were characterized by N_2_ adsorption and desorption using a Brunauer–Emmett–Teller analyzer (BET, Autosorb-iQ2, Quantachrome Instruments, Shanghai, China). The structures of the composite fibers were analyzed by FTIR spectroscopy (FTIR-4100, Jasco, Shanghai, China), within the wavenumber ranging from 400 to 4000 cm^−1^ and a wide angle X-ray diffractometer (XRD, D/Max, Rigaku, Beijing, China), under mode of 40 kV and 50 mA, λ = 1.5406 Å radiation using copper-K alpha. The samples were scanned in the 2θ range from 5° to 60° with a scanning rate of 30 min^−1^. The chemical composition of membrane surfaces were analyzed by XPS (ESCALab220i-XL, VG Scientific, Waltham, MA, USA) with Al K_α_ X-ray source, Ni-filtered radiation, 40 kV of tube electric pressure and 30 mA of tube electric current.

### 2.5. Investigations on MO Adsorption and Photocatalysis Study

Organic dye MO was used as the adsorbate for adsorption and photocatalysis studies of fiber mats. 20 mg of fiber mats were immersed in 40 mg/L MO aqueous solution (20 mL). The solution with fiber mats was stirred in dark conditions for 1 h to reach the adsorption equilibrium. After that, the mix solution was irradiated for 30 min in a photodegradation reactor under UV light (wavelength of 365 nm) by mercury lamp (25 W). The mixed solution (6 mL) was taken out every 5 min and centrifuged. The centrifugation was used to separate the solid particles in the suspension from the liquid to obtain a supernatant for subsequent absorbance detection. The UV spectrophotometer of the residual MO content was recorded. The process was repeated six times. The other set of photocatalytic experiments were performed under visible light (wavelength of 600 nm) using a 350 W xenon lamp. The operation was the same as the process using UV light. The concentration of the residual MO was recorded every 20 min for 120 min. The adsorption and photocatalytic degradation R (%) of membranes to MO was calculated by the Equation (1):(1)$$R\left(\%\right)=\frac{A_{0}-A_{t}}{A_{0}}\times 100\%$$
where *A*_0_ and *A_t_* are the original and equilibrium absorbances of the MO solutions.

The batch experiments of MO onto the fiber membranes were carried out as functions of TiO_2_ concentration (0–5 wt %), pH (4–9), contact time (0–85 min), and initial concentration (10–80 mg/L) at room temperature. The adsorption and photocatalysis ability was computed as follows:(2)$$q_{t}=\frac{C_{0}-C_{t}}{m}\times V$$
where *q_t_* (mg/g) represented the removal amount at time *t*, *C*_0_ (mg/L) and *C_t_* (mg/L) are the original and equilibrium concentrations of the MO solutions, *V* (L) is the volume of MO solutions, *m* (g) is the quantity of fiber membrane.

In the reusability experiment, the used adsorbent was washed by deionized water. Then the supernatant of the solution, after the reaction was replaced by fresh MO solution. The photodegradation reaction was carried out under UV illumination with magnetic stirring for 1 h. This experiment was cycled five times.

## 3. Results and Discussion

### 3.1. Morphology Observation

The morphology and diameter distributions of the CA substrate are displayed in Figure 2. The surfaces of the CA fibers were smooth without beads. The mean diameter of CA fibers was 0.55 ± 0.24 μm (shown in Figure 2b).

The morphology and the diameter distributions of the CS microspheres with different additions of TiO_2_ are shown in Figure 3. Pure CS showed concave hemisphere morphology as shown in Figure 3a. With the content of 1, 2, 3 wt % TiO_2_, CS still maintained the concave hemisphere morphology, and TiO_2_ nanoparticles were evenly distributed on the surface of CS hemispheres (see Figure 3b–d). When TiO_2_ content was 4 wt % and 5 wt %, TiO_2_ nanoparticles begun to accumulate, and the original uniform morphology of CS hemispheres gradually disappeared (Figure 2e,f).

The particle size of pure CS hemispheres was 1.07 ± 0.24 μm. With the increase of TiO_2_ content from 1 wt % to 3 wt % (Figure 3b_1_–d_1_), the diameters of CS hemispheres were 1.35 ± 0.26 μm, 1.40 ± 0.28 μm and 1.46 ± 0.34 μm, showing an upward trend. This was because the increase of viscosity was more important than the conductivity of the electrosprayed solution (Table 1). As the content of TiO_2_ continued to increase, the diameters of CS hemispheres were 1.10 ± 0.24 μm and 1.11 ± 0.30 μm, respectively. The quick increase in the viscosity of the solutions led to the agglomeration of TiO_2_ on the surface of CS hemispheres, as shown in Figure 3e_1_,f_1_. The size of the decrease in the CS hemispheres was due to an increase in the conductivity of the electrospray solution.

### 3.2. XRD Characterization

The XRD patterns of all the samples are illuminated in Figure 4. The broad peak at 20.10° and 22.01° were characteristic peaks of CS and CA, respectively [47,48]. TiO_2_ powder showed three sharp feature peaks at 48.06°, 37.86° and 25.41° of 2θ values [45]. Compared with XRD of CA powders, the peak at 22.01° of CS/CA was weakened due to the addition of amorphous CS which inhibited crystallization of CA. Compared with XRD of CS and CA powders, the peaks at 20.10° and 22.01° of C-T-3/CA were weakened due to the addition of TiO_2_ which inhibited crystallization of CS and CA. It indicated that there were interactions between TiO_2_ and CS/CA, which might stabilize TiO_2_ on the membranes [45].

### 3.3. MO Removal Analysis

#### 3.3.1. Effect of TiO_2_ Content on MO Removal

The effect of the amount of TiO_2_ on the removal of MO under visible light is demonstrated in Figure 5. Before visible light irradiation, the absorbance of MO solutions for all samples decreased due to adsorption of MO by the CS based membrane. Under visible light, the absorbance of all solutions with different fibrous membranes decreased as the reaction time increased (shown in Figure 5a). There was a significantly faster removal trend for the MO in visible light than in darkness because of the effects of both adsorption and photocatalysis. The removal of MO consisted of two stages for all samples. In the first stage, as the reaction time increased, the removal rate of MO was fast. In the second stage, the removal rate declined and got close to an equilibrium state. All samples could reach saturation state at 120 min. In combination with Figure 5b, the removal amount increased at first and then decreased with a further increase of TiO_2_ content. When the TiO_2_ content was higher than 3 wt %, the MO removal amount declined. This might be attributed to the reduction of the reaction sites by the TiO_2_ agglomeration. Therefore, the best adsorption and photocatalysis performance was obtained at 3 wt % content of TiO_2_ for CS based adsorbents.

The effect of TiO_2_ contents on the MO removal under UV light is demonstrated in Figure 6. Under UV light, the absorbance of all solutions decreased faster than samples in darkness because of concurrent adsorption and photocatalytic degradation as the reaction time increased (shown in Figure 6a). A similar situation was found as in the visible light, where the removal of MO consisted of two stages for all samples. In the first stage, the removal rate of MO was fast as reaction time increased. In the second stage, the removal rate declined and then got close to an equilibrium state. All samples could reach saturation state at 30 min [20]. In Figure 6b, it can be seen that the removal amount increased at first and then decreased with further increase of TiO_2_ content. When the TiO_2_ content was higher than 3 wt %, the MO removal amount declined. This might be attributed to the reduction of the reaction sites by the TiO_2_ agglomeration. The removal amount of MO by adsorption and photocatalytic degradation under UV light were superior to that in visible light because of more energy for electronic transitions [23]. Therefore, C-T-3/CA was selected for further MO removal studies by UV irradiation in the following tests.

#### 3.3.2. Effect of pH Values on the MO Removal

The pH value, as a crucial parameter, controlled the MO removal ability by affecting the degree of ionization of hydrogen ions, hydroxide ions in solutions and structural stability. Figure 7 shows the removal amount of MO in the pH value range from 4 to 9 under UV light. Both samples exhibited high removal amount with pH value in the range of 4–6, and MO removal amount gradually declined as the pH value increased. C-T-3/CA had a higher removal amount than CS/CA due to both adsorption and photocatalysis for C-T-3/CA. The removal amount was greatly improved by the addition of TiO_2_ (that allowed for simultaneous adsorption and photocatalysis) under acidic conditions. This was because under acidic conditions, the –NH_2_ on the CS molecule was protonated, and TiO_2_ existed in the form of TiOH_2_^+^, resulting in the electrostatic attraction among—SO_3_^−^ in MO, –NH_3_^+^ and TiOH_2_^+^ [48,49]. Under alkaline conditions, the –NH_2_ on the CS molecule was deprotonated, TiO_2_ generated the TiO^−^, and an electrostatic repulsion appeared between fibrous membranes and MO, which resulted in the decrease of removal amount [47]. The maximum removal amount of MO for both samples was obtained at pH = 4.

#### 3.3.3. Effect of Initial MO Concentration on the MO Removal

The effect of initial MO concentration on MO removal is indicated in Figure 8. The MO removal ratio of CS/CA and C-T-3/CA decreased with the increase of the initial concentration of MO. At a lower initial concentration of MO, fiber membranes CS/CA and C-T-3/CA provided sufficient reactive sites for removal of MO. With the increase of initial concentration of MO, the total amount of MO molecules increased. However, the amount of fiber membranes remained unchanged, meaning that the amount of reactive sites did not change. Therefore, the removal ratio of MO decreased at the higher initial concentration of MO. C-T-3/CA showed a higher removal ratio of 98.4% than CS/CA at initial concentration of 10 mg/L due to simultaneous adsorption and photocatalysis for the MO removal by C-T-3/CA.

#### 3.3.4. Effect of Contact Time and Kinetic Study

The effect of time on the removal capacity of MO by CS/CA and C-T-3/CA are shown in Figure 9a. As time increased, the removal amount of MO by CS/CA and C-T-3/CA increased and then gradually reached the equilibrium state. In the initial stage (*t* < 25 min), due to the large number of active sites in both adsorbents, the removal amount of MO increased rapidly with the increase of contact time. The removal amount of MO increased slowly and finally reached the equilibrium 25 min later [49]. For CS/CA, the adsorption equilibrium appeared at 40 min, while for C-T-3/CA, equilibrium was reached at 60 min. Due to the addition of TiO_2_ nanoparticles, which increased the specific surface area of the fiber membrane, more time was token for the MO to penetrate the fiber membranes. Therefore, the contact time of 60 min was used for subsequent tests to ensure sufficient adsorption and photocatalytic degradation treatment. The pseudo-first-order and pseudo-second-order kinetic models were used to study the behavior of MO removal of fiber membranes [46], which can be expressed as follows:

Pseudo-first-order kinetic model:(3)$$\log\left(q_{e}-q_{t}\right)=\log q_{e}-\frac{k_{1}}{2.303}t$$

Pseudo-second-order kinetic model:(4)$$\frac{t}{q_{t}}=\frac{1}{k_{2}q_{e}^{2}}+\frac{t}{q_{e}}$$
where *q_e_* and *q_t_* (mg/g) represent the equilibrium removal amount and the removal amount at time *t*; *k*_1_ and *k*_2_ are the constants pseudo-first-order and pseudo-second-order rate, respectively.

The experimental kinetic curves of MO removal are shown in Figure 9b,c. The relevant kinetic parameters calculated from the curves are listed in Table 2. The correlation coefficients of the pseudo-first-order model and the pseudo-second-order model for MO removal by CS/CA were nearly equal to and greater than 0.95, indicating that physical adsorption and chemical adsorption all contributed to the removal of MO. The *R*^2^ of the pseudo-second-order model (0.9952) was higher than that of the pseudo-first-order model (0.9878) for the removal of MO by C-T-3/CA. This suggested that fiber membranes exhibited mainly chemical adsorption accompanied by physical adsorption via the addition of TiO_2_ [50].

#### 3.3.5. Isotherm Study

The Langmuir model and Freundlich model were used to describe the isotherm equilibrium data of MO removal by CS/CA and C-T-3/CA fiber membranes [50]. It can be expressed as Equations (5) and (6)

The Langmuir model:(5)$$\frac{C_{e}}{q_{e}}=\frac{1}{q_{m}K_{L}}+\frac{C_{e}}{q_{m}}$$

The Freundlich model:(6)$$\log q_{e}=\frac{1}{n}\log C_{e}+\log K_{F}$$
where *q_e_* (mg/g) represents the equilibrium removal amount, *C_e_* (mg/L) is the equilibrium concentration of MO in aqueous solution and *q_m_* (mg/g) represents the maximum removal amount, *K_L_* is the Langmuir isotherm constant related to free energy and *K_F_* and *n* are the Freundlich isotherm constant which expressed removal capacity and removal intensity, respectively.

The experimental isotherm curves of MO removal are shown in Figure 10a,b. The relevant isotherm parameters calculated from the curves are listed in Table 3. On the basis of the correlation coefficients of MO removal by CS/CA, the Freundlich isotherm model (Figure 9b) was better fitted than the Langmuir isotherm model (Figure 9a). Therefore, the CS/CA mainly exhibited multiple layers of adsorption. After the addition of TiO_2_, the correlation coefficients of the Langmuir model and Freundlich model were approximately equal, indicating that C-T-3/CA exhibited multiadsorption behavior. This might be due to both the effects of photocatalytic degradation and adsorption [50].

### 3.4. Removal Mechanism Analysis

#### 3.4.1. FTIR Spectra Results

The FTIR spectra of C-T-3/CA fibers before and after removal of MO are demonstrated in Figure 11. Before removal of MO, the peaks at 3100–3500 cm^−1^ were ascribed to –OH and –NH_2_, 1638 and 1572 cm^−1^ were attributed to –NH_3_^+^ and –NH_2_ of CA and CS, respectively. The peak appearing at 1154 cm^−1^ was attributed to the stretching of the C–O–C band of CA. The peaks in the range of 400–800 cm^−1^ were ascribed to the characteristic absorption of Ti–O [47,51]. After the adsorption and photocatalytic degradation of MO, the peak of the –NH_2_ group at 1572 cm^−1^ decreased and the –NH_3_^+^ at 1638 cm^−1^ increased [49]. It demonstrated that –NH_2_ was protonated as –NH_3_^+^ in the process of MO removal. Several new characteristic peaks that appeared were listed as follows: 1400–1450 cm^−1^ were ascribed to –N=N– group, and 1039 cm^−1^ was attributed to –SO_3_^−^ of MO. It showed that MO was adsorbed on surfaces of CS molecules during the reaction process. The peaks at 3100–3500 cm^−1^ increased, which might be due to the formation of abundant active hydroxyl radicals (·OH) during the photodegradation process (see Equations (9) and (10)).

#### 3.4.2. XPS Spectra Results

As shown in Figure 12, XPS was employed to analyze the chemical element of the membrane surface before and after the removal (that allowed for coinstantaneous adsorption and photocatalysis) of MO with full-range and magnified spectra of S, N, O and Ti. It can be seen in Figure 12a that C-T-3/CA membranes before MO removal showed peaks of binding energy at 285.5, 397.7, 531.2 and 457.1 eV, which were assigned to carbon atom (C 1s), oxygen atom (O 1s), nitrogen atom (N 1s) and titanium atom (Ti 2p), respectively. As seen in Figure 12b, the peak of S 2p appeared at 167.7 eV after removal of MO, indicating MO adsorbed on the surface fiber membranes. For the N 1s spectrum in Figure 12c, the peak at 399.1 eV was assigned to –NH_2_ group before MO removal. After MO removal, a new peak appeared at 401.7 eV and was attributed to –NH_3_^+^, and the peak of –NH_2_ at 399.1 eV shifted to 399.8 eV. This indicated that –NH_2_ was protonated as –NH_3_^+^, and electron transfer occurred between –NH_2_ and MO molecules, respectively [22]. As seen in Figure 12d,e, the O 1s and Ti 2p binding energy increased after the removal of MO. Ti could exist in the form of TiOH_2_^+^ in acid solutions. After the reaction with MO molecules, the groups of –OH_2_^+^ as electron acceptors could interact with –SO_3_^−^ of MO due to electrostatic attraction, which could influence charge density of Ti and O [20].

#### 3.4.3. BET Analysis

The BET results of CS/CA and C-T-3/CA membranes were measured by nitrogen adsorption method (see Table 4). After the addition of TiO_2_, the fiber membrane attained higher specific surface area and pore volume owing to uniform distribution of TiO_2_ on CS hemispheres and an increase of the distance among C-T hemispheres (see Figure 3d). Therefore, the relative high adsorption capacity of MO on C-T-3/CA was ascribed to the high specific surface area, which should be good for the photodegradation of MO.

#### 3.4.4. Mechanism Analysis

According to the above analysis, the high removal capacity of C-T-3/CA was mainly due to the effective dispersion of TiO_2_ in combination with CS, which improved the specific surface area of fiber membranes and increased the sites of photocatalytic degradation and adsorption. MO molecules could be adsorbed to the vicinity of photodegradation sites by electrostatic attraction between CS and MO molecules, promoting the contact probability of TiO_2_ and MO in the following photocatalytic degradation process. The reaction mechanism between MO and C-T-3/CA is illustrated in Figure 13. The mechanism of photocatalysis is summarized from Equation (7) to Equation (12). The MO removal mechanism included: (1) Adsorption, the electrostatic attraction between –NH_3_^+^ group of CS and –SO_3_^−^ of MO molecules; (2) photocatalysis, the photocatalytic degradation of the MO by UV illumination began with photoexcitation of the TiO_2_ and then formed an electron-hole pair (Equation (7)). High oxidation valence-band holes (h^+^) directly oxidized MO degradation (Equation (8)). Water decomposition produced ·OH (Equation (9)) or a reaction of h^+^ with OH^−^ (Equation (10)). Meanwhile, the reaction between conduction-band electrons (e^−^) and proper electron acceptors (such as O_2_) yielded oxidative radicals as described by Equation (11). The generated hydroxyl radicals easily degraded MO to form inorganic small molecules (Equation (12)) [6]. These results agreed with BET, FTIR and XPS analyses.
(7)$${\mathrm{TiO}}_{2}+\mathrm{hv}\to e^{-}+h^{+\;}$$
(8)$$h^{+}+\mathrm{MO}\to {\mathrm{MO}}^{\cdot +}\to \mathrm{Oxidation}\;\mathrm{of}\;\mathrm{MO}$$
(9)$$h^{+}+H_{2}O\to H^{+}+\;\cdot \mathrm{OH}$$
(10)$$h^{+}+{\mathrm{OH}}^{-}\to \;\cdot \mathrm{OH}$$
(11)$$e^{-}+O_{2}\to \;\cdot O_{2}^{-}$$
(12)$$\mathrm{MO}+\cdot \mathrm{OH}/\cdot O_{2}^{-}\to \mathrm{Degradation}\;\mathrm{of}\;\mathrm{MO}$$


### 3.5. Recycling Ability

To research the recycling ability of the composite membranes, the absorbance changes were measured in five cycles of MO removal. As demonstrated in Figure 14, the MO removal of fiber membranes was almost invariable for five consecutive cycles. The membranes still had a removal ratio of 98% at the end of the fifth cycle, testifying that the membranes maintained good performance of adsorption and photocatalytic degradation. This was because the stability of CS in aqueous solutions was improved by the intermolecular hydrogen bonds between CS and CA, and TiO_2_ could be uniformly distributed via the interactions with CS molecules for effective photocatalytic degradation of MO [45].

## 4. Conclusions

In this work, the C-T/CA fiber membranes were prepared via electrospinning and electrospraying. After the addition of TiO_2_, the removal capacity of MO was improved, as a result of concurrent adsorption and photocatalytic degradation. When TiO_2_ content was 3 wt %, the highest MO removal amount for fiber membranes (C-T-3/CA) was reached, 98% at pH value of 4 and MO concentration of 40 mg/L. The kinetic and isotherm model for the removal of MO by fiber membranes were well fitted pseudo-second-order kinetic and Freundlich isotherm models. Especially, C-T-3/CA membranes exhibited multiple layer adsorption with the addition of TiO_2_. The involved mechanisms including adsorption and photocatalytic reaction occurred in the removal process which was demonstrated by FTIR, XPS and BET results. The recycling experiment showed fiber membrane had excellent stability and reusability.

## Figures and Tables

**Figure 1 polymers-11-01293-f001:**
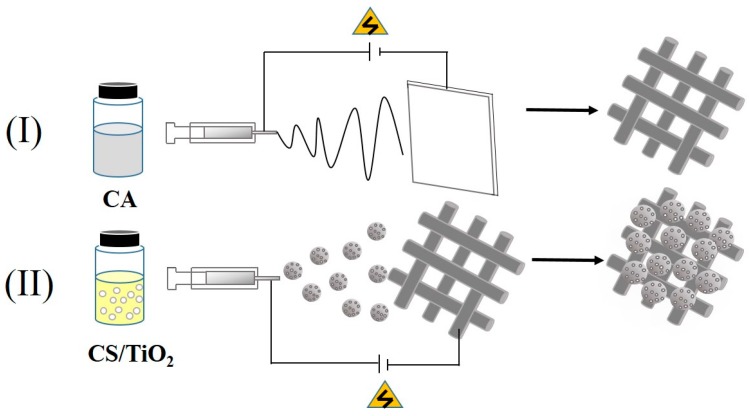
The preparation of C-T/CA fiber membranes: (**I**) Electrospinning, and (**II**) electrospraying.

**Figure 2 polymers-11-01293-f002:**
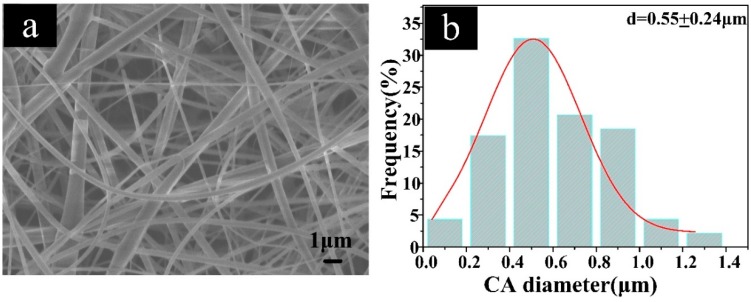
(**a**) The morphology, and (**b**) the diameter distribution of the CA fibers.

**Figure 3 polymers-11-01293-f003:**
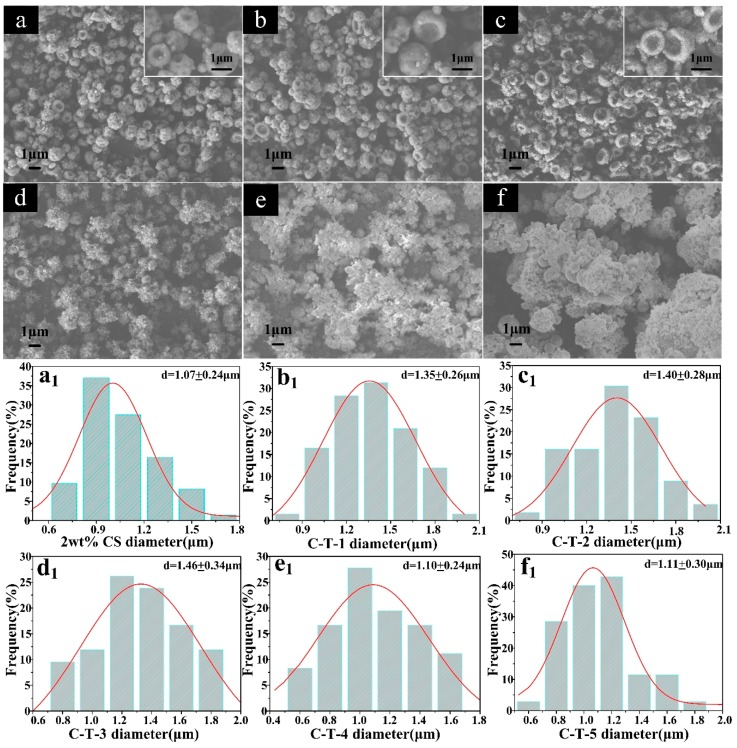
SEM images and diameter distributions of the microspheres. (**a**) and (**a_1_**) 2 wt % CS, (**b**) and (**b_1_**) C-T-1, (**c**) and (**c_1_**) C-T-2, (**d**) and (**d_1_**) C-T-3, (**e**) and (**e_1_**) C-T-4, (**f**) and (**f_1_**) C-T-5.

**Figure 4 polymers-11-01293-f004:**
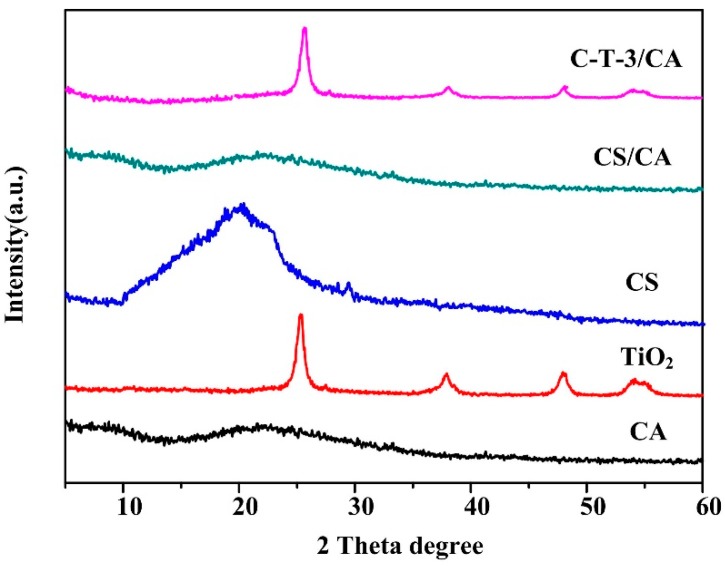
XRD spectra of CA, TiO_2_, CS powders, CS/CA and C-T-3/CA fiber membranes.

**Figure 5 polymers-11-01293-f005:**
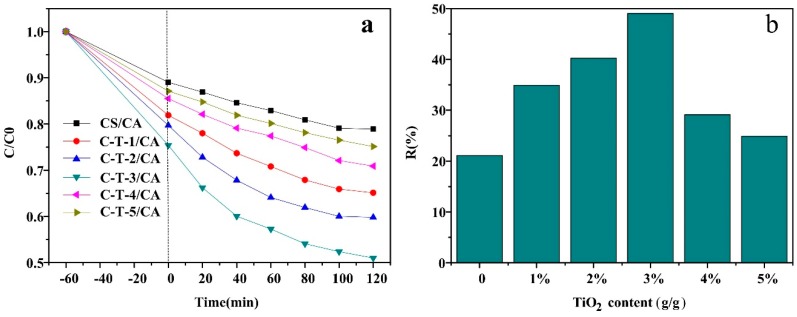
MO removal performance under visible light. (**a**) The effect of reaction time on absorption amount, (**b**) the effect TiO_2_ content on removal rate.

**Figure 6 polymers-11-01293-f006:**
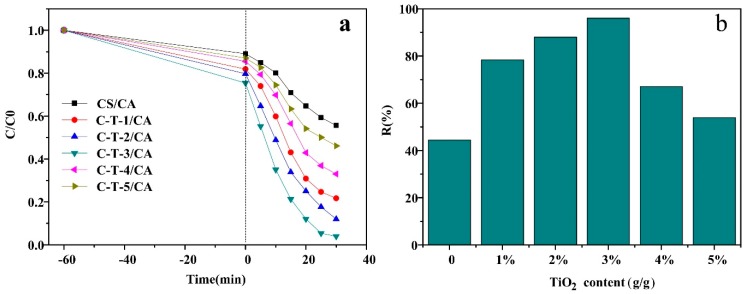
MO removal performance under UV light. (**a**) The effect of reaction time on absorption amount. (**b**) The effect of TiO_2_ content on removal rate.

**Figure 7 polymers-11-01293-f007:**
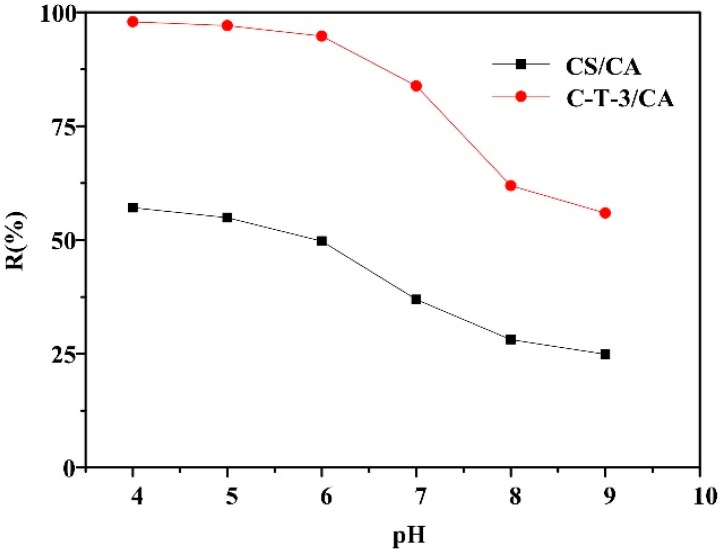
Effect of pH values on the MO removal amount (initial concentration 40 mg/L).

**Figure 8 polymers-11-01293-f008:**
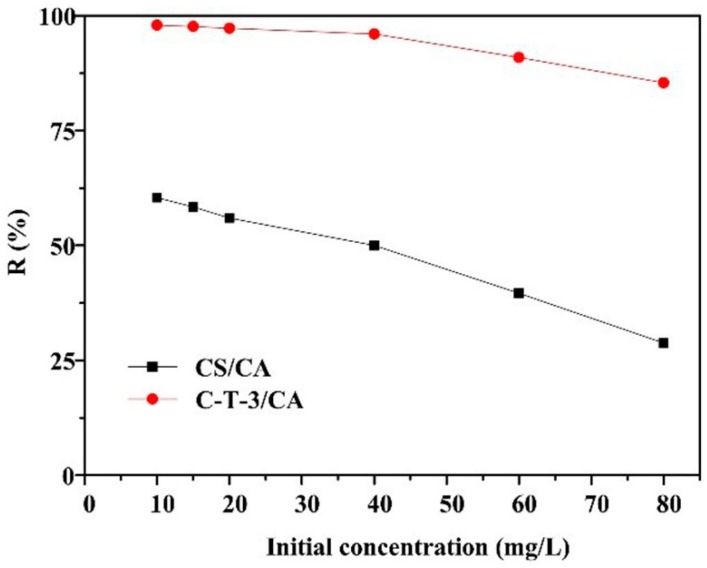
Effect of initial concentration on the MO removal amount.

**Figure 9 polymers-11-01293-f009:**
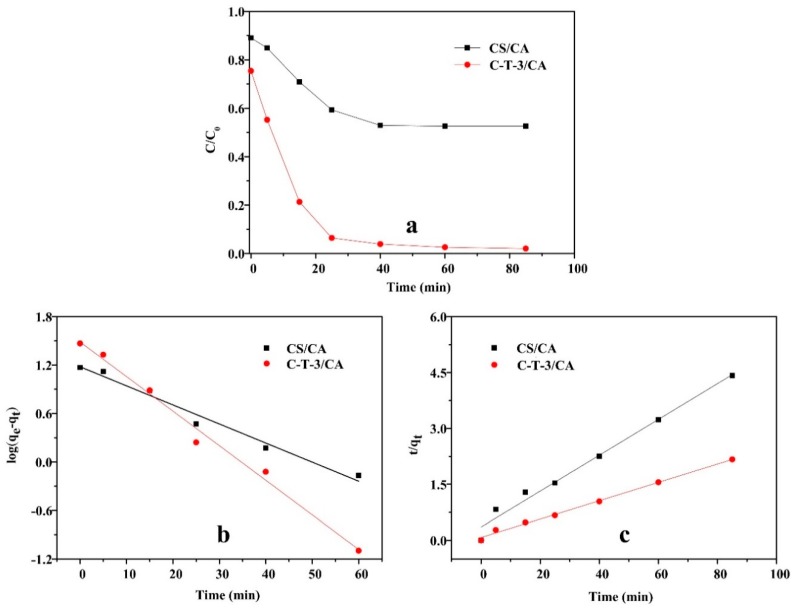
Effect of contact time on the MO removal amount. (**a**) The kinetics model for MO removal, (**b**) pseudo-first-order model, and (**c**) pseudo-second-order model.

**Figure 10 polymers-11-01293-f010:**
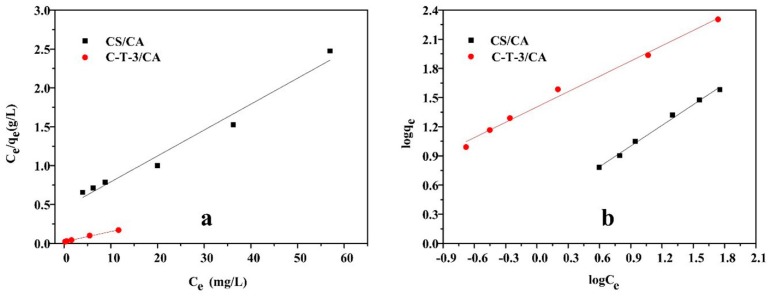
Isotherm model of MO removal: (**a**) Langmuir model, (**b**) Freundlich model.

**Figure 11 polymers-11-01293-f011:**
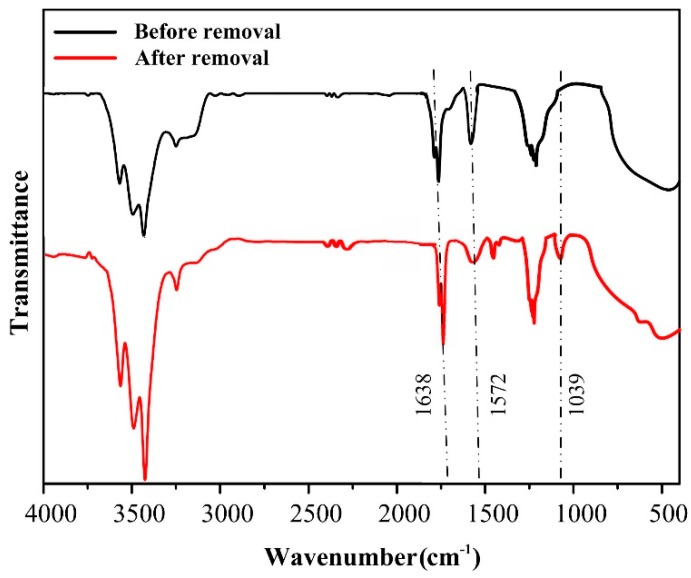
FTIR spectra of C-T-3/CA before and after MO removal.

**Figure 12 polymers-11-01293-f012:**
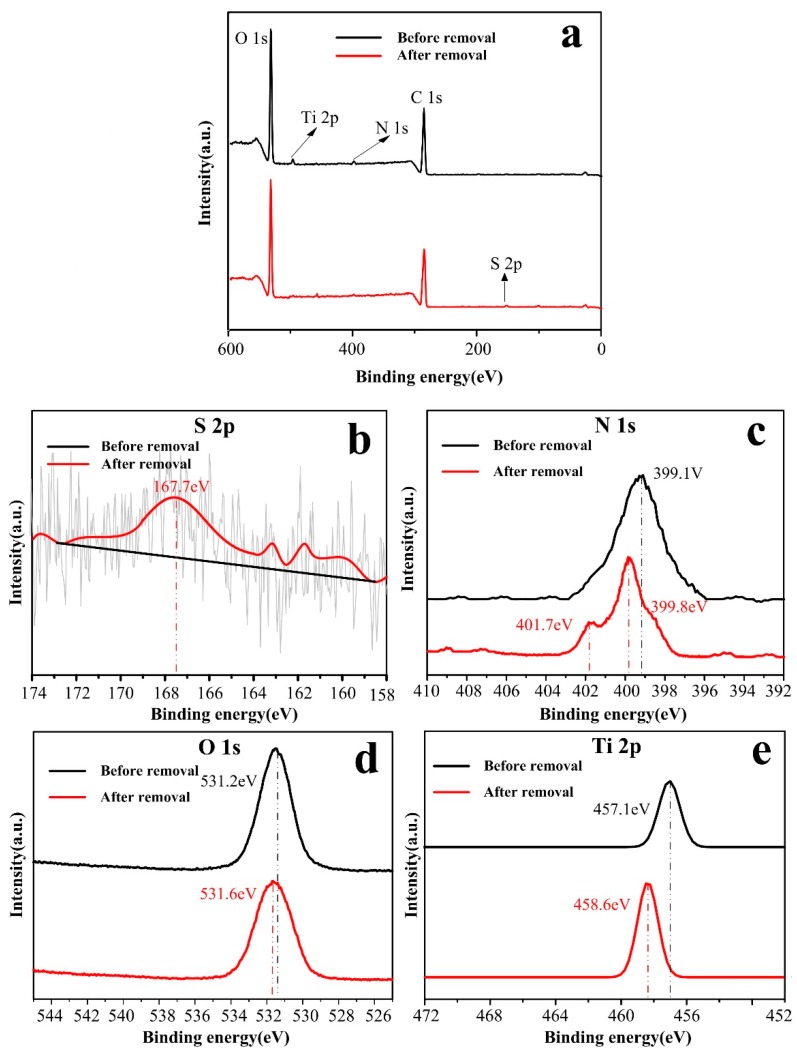
XPS spectra of C-T-3/CA before and after MO removal.

**Figure 13 polymers-11-01293-f013:**
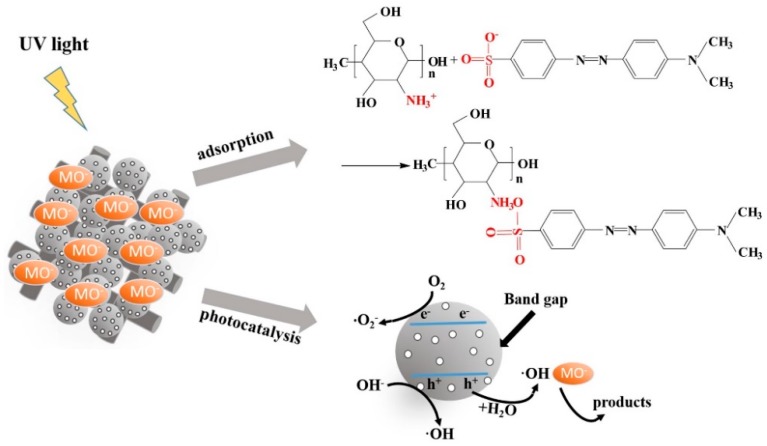
The mechanism of MO removal by C-T-3/CA.

**Figure 14 polymers-11-01293-f014:**
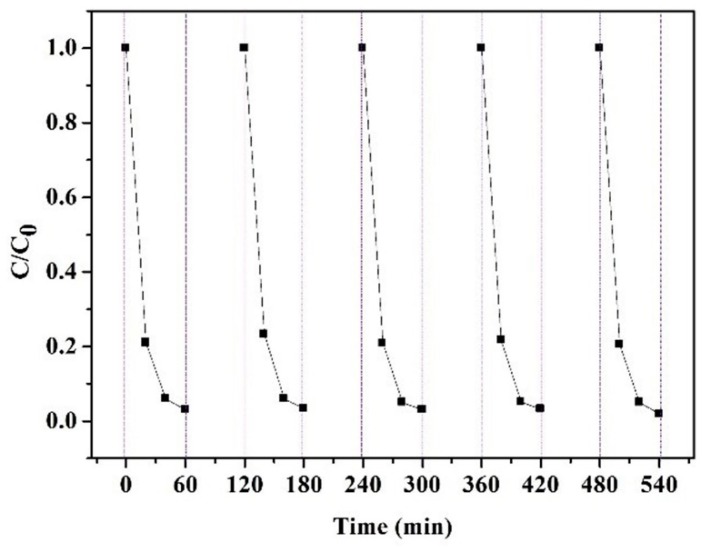
MO removal by C-T-3/CA in five cycles.

**Table 1 polymers-11-01293-t001:** The parameters of C-T-X electrospraying solutions.

TiO_2_ Content	0%	1%	2%	3%	4%	5%
**Viscosity (mPa s)**	140.7	158.8	172.5	183.6	209.5	220.4
**Conductivity (μS cm^−1^)**	219.0	230.7	243.8	256.1	265.0	273.4

**Table 2 polymers-11-01293-t002:** Kinetics parameters for MO removal.

Sample	*q_e_*, *exp* (mg/g)	Pseudo-First-Order Model			Pseudo-Second-Order Model		
		*k* _1_	*R* ^2^	*q_e_*, *cal* (mg/g)	*k* _2_	*R* ^2^	*q_e_*, *cal* (mg/g)
CS/CA	19.24	5.44 × 10^−2^	0.9748	15.08	13.48 × 10^−2^	0.9799	20.80
C-T-3/CA	39.20	9.86 × 10^−2^	0.9878	30.61	32.33 × 10^−2^	0.9952	40.67

**Table 3 polymers-11-01293-t003:** Isotherm parameters for MO removal.

Sample	Langmuir Model			Freundlich Model		
	*K_L_*	*q_m_* (mg/g)	*R* ^2^	*K_F_*	*n*	*R* ^2^
CS/CA	7.28 × 10^−2^	30.01	0.9714	2.30	1.41	0.9932
C-T-3/CA	62.5 × 10^−2^	76.22	0.9948	25.33	1.90	0.9997

**Table 4 polymers-11-01293-t004:** BET results of CS/CA and C-T-3/CA.

Sample	Specific Surface Area (m^2^/g)	Pore Volume (cc/g)	Average Pore Diameter (nm)
CS/CA	301.30	0.33	4.39
C-T-3/CA	342.10	0.48	4.79

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
