# Peer review of "TiO2-Doped Chitosan Microspheres Supported on Cellulose Acetate Fibers for Adsorption and Photocatalytic Degradation of Methyl Orange"

_polymers, 2019, doi:10.3390/polym11081293_

Reviewer 1 Report

This paper reports on adsorption and photocatalytic degradation of methyl orange by TiO2-doped Chitosan microspheres supported on Cellulose acetate fibers. This manuscript is well characterized the samples with various suitable analysis methods and contains valuable scientific information. . So, I believe this paper can be considered for publication in Polymers after minor revision. Suggestions for revision are given below:

In the XRD characterization section, authors explained that the characteristic peak of C-T-3/Ca at 25.4o was slightly shifted to the right compared with TiO2 powders due to interactions between –OH of CS molecules and Ti atom. Is your meaning such interaction can be change the unit cell parameters of anatase TiO2? Another words, is this kinds of interaction make shortened the bond length of Ti-O in TiO2 crystalline?

In the XPS spectra results section, authors described that the increase of Ti 2p binding energy might be because of the formation of electron-hole pairs on TiO2 molecules under UV light. Did authors measure the XPS under UV light? As you know that electron-hole pairs can be generated under UV light but they are quickly recombined without UV irradiation. Is there any possibility for the change of oxidation state of Ti atom after removal of MO?

Authors should consider using clearer all of diagrams and figures. All the provided pictures have the low resolution.

In the table 4, please check the significant figures for the specific surface area and average pore diameter.

In line 123, author should write which wavelength they cut-off to make sure irradiated light from xenon lamp is visible light.

Write the full name of BET (Brunauer-Emmett-Teller) when you mention first.

Equation 2 (line 132), authors should define qt

Some typo errors: -

-Grammar needs polishing throughout the whole manuscript

-11. In line 134, m (g) is quantity, not quality.

-Line 69 electrspraying → electrospraying

-Line 104 and 121 token → taken

-In table 4, Special surface area à Specific surface area

-Line 332 and 333, double space

Reviewer 2 Report

The abstract needs a complete re-write.

What is the problem you are trying to solve and why is it worthwhile? What is the challenge (eg. fixing TiO2 in a matrix, recovery of the MO from the filter)? How do you solve it? Does it work? What does it mean for the problem you are trying to solve?

These are minimum questions that need to be answered in the abstract -be quantitative (how much MO absorbed/rejection rate from a certain volume of solution?). Giving investigation techniques is not always relevant in an abstract…you did SEM, so what?

All figure captions are simply figure titles. Good figure captions are descriptive of the science they convey and the main conclusion. Please modify them; refer to helpful guides, such as ‘How to write a paper’ by Mike Ashby, for example.

P1 lines 33-35 – verb missing. “It evoked problems..” this, and many other expressions are not helping the readability of the document and seem to have been put together with an online translator. Please run the text by a native English speaker before resubmission.

P2 line 90. What power for ultrasonication?

P3 line 104 ‘taken’, not token

P3 line 121 ‘taken’

P3 line 121. Why the need for centrifugation? Please quote centrifugation in g, rather than rpm

P3 lines 121 + 123: quote UV wavelengths or at least ‘short’ ‘long’

P4 line 154 quote 1.46 ± 0.34 as 1.5 ± 0.3 (and do the same for all other diameters quoted in this section)

P6 lines 171-172: The shift might be due to interactions between OH of CS molecules and Ti atom [45]. So what? What is the significance of this observation. Is this beneficial to the photocatalysis process? If there is no link, what is the point of doing XRD? There is nothing in the processing that could conceivably change the structure of the TiO2; perhaps only after UV irradiation.

P6 line 185  was consisted= consisted (and on the next page, check the whole manuscript please)

P7 Line 213 ‘controlling the degree of ionization” specify the ionization of what?

The kinetic models and the Langmuir and Freundlich models need referencing.

Fig 12 a) zoom only 600eV to 0eV. The rest is redundant; it will help convince the reader that the very noisy spectrum shown with higher energy resolution in 12 b) is believable, despite the massive noise in the spectrum. If using a filter, please describe the filter (eg Fourier cut-off – Nyquist frequency). I would normally expect a statement as to the SNR and why the peak is not ‘noise’, but agree that there is evidence of S.

Overall, these are minor corrections; the science is thorough and sound, just some of the language needs more careful translation and the abstract would need a full re-write.
